# Psychological stress and associated factors among municipal solid waste collectors in Hanoi, Vietnam: A cross-sectional study

**Quynh Thuy Nguyen**[1], **Bang Van Nguyen**[2]*, **Ha Thi Thu Do**[3], **Bich Ngoc Nguyen**[1], **Van Thanh Nguyen**[4], **Son Thai Vu**[1], **Thuy Thi Thu Tran**[1]

**1** Department of Occupational Health and Safety, Hanoi University of Public Health, Hanoi, Vietnam,
**2** Department of Hematology, Toxicology Radiation and Occupational Diseases, Vietnam Military Medical University, Hanoi, Vietnam, **3** Cau Giay District Health Centre, Hanoi, Vietnam, **4** Hanoi University of Public Health, Hanoi, Vietnam

* bsbanga7bv103@gmail.com

## Abstract

### Introduction

In low and middle-income countries, the manually operated municipal waste collection system prominently depended on the performance of waste collectors (WC). Most of the literature has focused on the impact of waste collection tasks on WCs' physical health, while little was known about the psychological effects of work-related stress. This study aimed to examine the prevalence of psychological stress and related factors among waste collectors in Hanoi, Vietnam.

### Methods

A cross-sectional survey was conducted among 802 WCs in 2017. The questionnaire included the 7-item Stress component of the 21-item Depression, Anxiety and Stress Scale, and questions on demographics and work conditions. Descriptive and multivariate logistics regression analyses were conducted to examine the factors related to psychological stress among WCs.

### Results

Results showed that 13.4% of WCs reported stress symptoms; among them, 3.3% of WCs experienced severe stress. Factors related to lower odds of self-reported psychological stress included self-perceived frequent exposure to high and low temperatures in the working environment (OR = 0.51 and 0.52, respectively). Factors associated with the increased likelihood of symptoms included frequent exposure to hot/flammable objects (OR = 2.41), working a night shift in the last three months (OR = 1.82), education lever lower than high school (OR = 1.82), and having an insufficient monthly income (OR = 1.99).

**Data Availability Statement:** The minimal dataset is provided as Supporting information of this paper.

**Funding:** The Hanoi Department of Science and Technology provided monetary fund for Quynh Nguyen Thuy to implement the original study on which the present article is based. The funders had no role in study design, data collection and analysis, decision to publish, or preparation of the manuscript.

**Competing interests:** The authors have declared that no competing interests exist.

## Conclusion

The high percentage of workers with severe stress implies the need for mental health prevention and treatment for WCs who participated in this study.

## Introduction

Municipal solid waste (MSW) is a problem in developing countries [1, 2]. The problem is expected to become more significant with their rapid urbanization and rate of population growth that exceeds the capacity of current infrastructure, especially in metropolises with a high density of residences. In addition, the incompatibility between the application of technology in the MSW management process and contemporary urban planning results in the continued use of manual handling and wheeled containers as the main method to collect MSW in narrow alleys and dead ends [2–4]. Waste collectors (WC) have been and will be an irreplaceable component of cities' MSW management in developing countries for years to come. The quality and efficiency of MSW systems, therefore, considerably depend on the performance and well-being of the workforce.

However, MSW collection is a dangerous job [2]. Studies have reported a high prevalence of musculoskeletal disorders, burnout, and fatigue among WCs [5, 6]. The job is primarily done during the night shift (usually from 6 p.m. to 2 p.m. on the next day) when the road is free of traffic [4]. The work schedule considerably affects workers' circadian rhythms, leading to a high risk of later sleep deprivation and disorders and work-home conflict which, consequently, intensifies the impact of psychological and physical job demands on night workers [4]. WCs also are highly susceptible to work-related injuries such as traffic accidents, slipping, falling, and injury by sharp objects [7, 8]. Despite all the hard work and danger of the job, insufficient income for workers to support themselves and their family, in association with the social pressure of a "low status" job [9, 10], may increase insecurity, anxiety and psychological stress among WCs.

In our review of the literature, we found many reports of a variety of physical health problems among workers in this occupation, but few publications describing workers' exposure to work-related stressors [10–12] or their harsh work environment [4]. Only one paper reported multiple indicators of mental health status among Japanese MSW workers in 2019, including tension/anxiety (45.5%), depression-dejection (20.0%), or social dysfunction (18.2%) [13]. However, Japan's waste management system is highly developed with integrated approaches and advanced waste disposal techniques to reduce waste generation rate and support WCs [1]. The improvement of work conditions for Japanese MSW workers has reduced exposure to occupational hazards and risks prevalent in lower-income countries to a remarkable extent [2, 14].

Similar to other countries in the world, a metropolis like Hanoi, the capital of Vietnam, also faces the challenge of sustainable MSW management [3, 15, 16]. Rapid urbanization, lifestyle changes, and high population density in the city generate an enormous amount of MSW every day, which is collected manually by WCs [17]. Since 2018, the population of Hanoi has increased to over 7.5 million people and the population density to 2.239 people/km$^2$ [18]; its residents resided in 30 district-level administrative units, towns, 584 communes, wards, and towns [19]. Hanoi is located in the tropical monsoon region; hence, the climate is divided into four distinct seasons throughout the year. Temperatures can reach 40°C in summer and below 5°C in winter [20].

According to the Hanoi Department of Natural Resources and Environment, in 2019, about 6,500 tons of domestic MSW was generated daily, of which the MSW of 12 urban districts and Son Tay town accounted for 53.9%, with a 99–100% collection rate. MSW in 17 suburban districts was less than half of the daily amount, but only 87 to 88% were appropriately collected for disposal [15]. The laborious workload, coupled with harsh working conditions such as extreme heat in the summer and low temperatures in the winter, in association with lack of mechanical support for heavy manual work, leads WCs to become physically and mentally exhausted. However, evidence regarding the risk of mental distress among WCs worldwide and particularly in Vietnam is limited. Therefore, this study aimed to examine the prevalence of psychological stress and related factors among WCs in Hanoi, Vietnam. Results from this study will contribute to evidence-based solutions for improving the well-being of WCs in Vietnam and countries with similar MSW systems.

## Materials and methods

### Study design and participants

MSW collection in Hanoi is mainly conducted by the urban environment one-member state-owned limited company (URENCO), the only state enterprise in waste management in Hanoi [16, 21, 22]. The company's affiliates include four companies in charge of MSW collection and transportation (URENCO 1, 2, 3, and 4), two companies involved in waste disposal and treatment (URENCO 6 and 7), and several subsidiaries [22].

This study, cross-sectional in design, was conducted in 2017 among workers for two of the four MSW collection companies, affiliates of URENCO, who were selected following cluster sampling. The study applied two inclusion criteria, namely (1) having a labor contract which was longer than three months with the current company and (2) collecting waste as a primary job responsibility by the time of recruitment. All eligible WCs (894) from the selected affiliates were invited to participate; 817 agreed to provide information. Fifteen questionnaires were removed because participants refused to provide information on critical variables. The final sample used for the analysis was 802 WCs (response rate of 89.7%).

### Data collection

Ten individuals, including researchers and graduate students with experience in conducting structured interviews, collected the data. They were trained thoroughly by the principal investigator on how to conduct the interview, and each practiced on five WCs who belonged to another waste collection company and did not participate in this survey. The principal investigator regularly monitored and supervised the interviewers' performance and progress directly on the WCs' worksite.

Data collectors performed face-to-face interviews at participants' worksite with permission of the company's management, using structured questionnaires. Informed consent forms were completed by the participants before data collection. In each worksite, data collectors contacted the team leaders of the WCs to confirm the dates of their visits, which were two to three times per month to facilitate the participation of as many WCs as possible.

The interviews were conducted before or after the WCs' work shift. Because most participants worked in the afternoon and at night and it was tough for them to read the questionnaires, data collectors read the questions aloud and recorded responses instead of the original self-administered method [23]. After the interview, participants received an herbal tea box as compensation for their participation regardless of their answers. The tea box was worth less than one USD.

## Measures

The survey questionnaires included two parts: Questions about psychological stress, and questions about potentially related factors.

Psychological stress was measured with the stress component of the Depression, Anxiety, and Depression scale 21 items (DASS-21), which was developed to measure symptoms of depression, anxiety, and stress [23]. This study only used the psychological stress component (DASS-S), which included seven 4-point items ranging from 0 (did not apply to me at all) to 3 (applied to me very much, or most of the time). The seven items measured the degree of chronic non-specific arousal. It assessed difficulty in relaxing, nervous arousal, and being easily upset or over-reactive and impatient. The sum of the seven item scores was the stress score (ranging from 0 to 21), which was then multiplied by 2 to create the final stress score (ranging from 0 to 42). Severity levels of psychological stress were categorized into normal, mild, moderate, severe, and extremely severe (with cut-off points of 15, 19, 26, and 34, respectively) [24]. The Vietnamese DASS 21 was previously reported to be reliable in measuring psychological problems among workers (i.e., hospital nurses) in Vietnam [25]. In the present study, a cut-off score of 15 was used to categorize participants into two groups: No stress (score less than 15) and With stress symptoms (score from 15 to 42). Cronbach's alpha coefficient for the DASS-S in this study was satisfactory (0.91).

Other factors related to stress among WCs included physical health, demographic characteristics, and study participants' working conditions.

Physical health was calculated using ten items from the physical functioning subscales of the SF-36 quality of life questionnaires (version 2), which measured the ability to perform daily activities such as running, lifting, climbing stairs, walking, etc. Scoring was a two-step process. First, precoded numeric values were recoded per the scoring key. Then, all item scores were converted so that a higher score defined a more favorable health state. Each item was scored on a 0 to 100 scale so that the lowest and highest possible ratings were 0 and 100, respectively. The physical function score represented the average for the ten items in the physical functioning subscales that the participants answered [26]. In this study, the physical functioning score did not have a normal distribution but had good internal consistency (Cronbach's alpha coefficient = 0.87). The median was selected as a cut-off point to divide participants into two groups, better physical health ($\geq$ 95) and worse physical health ($<$ 95).

Demographic information collected from the participants included age, gender, education level, marital status, number of children, monthly income, and years working as a WC (work seniority). Age and work seniority were categorized into two groups ($>$ 39 versus $\leq$ 39 years old and $\geq$ 15 versus $<$ 15 years of experience as a WC, respectively). Participants were asked their gender (male or female). Education level was classified into two groups, namely below high school and high school and above.

Work conditions were described in terms of two groups of variables: work organization and exposure to occupational hazards. Work organization variables included the number of work hours per shift ($\leq$ 8 versus 9–12 hours/shift) and work-shift during the last three months (Night shift, from 6 p.m. to 2 a.m., Others [including WCs working the day shift from 5 a.m. to 12 a.m., or the afternoon shift from 1 p.m. to 8 p.m.], or Frequently changed shifts during the last three months). Occupational hazards included self-reported frequency of exposure to different physical conditions (sunlight, hot/cold/wet weather conditions, noise, insufficient illumination), chemicals (toxic gas, dust, unpleasant smell), and other factors (inflammable, sharp object, threat of physical/mental violence). All variables were dichotomous (Frequent exposure versus Seasonal to No exposure, or Yes versus Never).

## Statistical analysis

The collected data were analyzed using SPSS Version 22.0. Descriptive analysis was done to assess the mean and standard deviation (SD) of the DASS-S scores and the prevalence of stress according to the level of severity. WCs were divided into two groups, no stress symptoms (DASS-S < 15) and the presence of stress symptoms or stress (DASS-S $\geq$ 15). Multiple logistic regression was conducted to examine factors associated with psychological stress (Enter method, significance level of $p < 0.05$).

## Ethical consideration

The study was approved by the ethics committee of biomedical research at the Hanoi University of Public Health, Hanoi, under Decision No. 46/2017/YTCC-HDD3, dated 15/02/2017. Participation in the study was completely voluntary, and written consent forms were obtained prior to data collection.

## Results

### Personal characteristics of waste collectors participating in this study

Table 1 showed that most participants were female workers (83.3%). The average age and years of services of WCs in this study were 40.1± 6.5 and 11.8 ± 7.1, respectively. More than 90% of WCs in this study were married, lived with their spouse (91.4%), and already had at least one child (97.5%). About 80% of WCs had completed high school. Three-fourth of WCs worked at night when the road was free of traffic (72.8%), and less than 20% of WCs reported that they had to work more than the official number of work hours as 8 hours per day (19.8%).

**Table 1. Characteristics of study participants (N = 802).**

| Variables | Subgroups | Whole sample | |
|---|---|---|---|
| | | N = 802 | % |
| Age (mean ± SD = 40.1± 6.5) | ≥39 | 321 | 40.0 |
| | <39 | 481 | 60.0 |
| Gender | Male | 134 | 16.7 |
| | Female | 668 | 83.3 |
| Marriage relationship | Others | 69 | 8.6 |
| | Married and live with a spouse | 733 | 91.4 |
| Children | Not yet | 20 | 2.5 |
| | ≥ 1 child | 782 | 97.5 |
| Education | Below high school | 155 | 19.3 |
| | High school and above | 647 | 80.7 |
| Years of service (mean ± SD = 11.8 ± 7.1) | ≥15 years | 205 | 25.6 |
| | < 15 years | 597 | 74.4 |
| Work shift in the last 3 months | Night shift | 584 | 72.8 |
| | Others | 218 | 27.2 |
| Work hours | > 8 hours/day | 159 | 19.8 |
| | ≤ 8 hours/day | 643 | 80.2 |
| Monthly income | Insufficient | 413 | 51.5 |
| | Normal to sufficient | 389 | 48.5 |
| Physical functioning (0–100) (mean ± SD = 90.4 ± 13.5) | < 95 | 490 | 61.1 |
| | ≥ 95 | 312 | 38.9 |

Generally, WCs had good physical health, with a physical functioning mean score of 90.4 out of 100, and about 40% of WCs had physical functioning scores over 95.

## Prevalence of psychological stress among waste collectors

The mean and standard deviation for the DASS-S were 9.7 and 7.8, respectively. Fig 1 showed that the prevalence of stress among WCs in this study was 13.4%. Among WCs reporting self-perceived stress, only 6% had mild symptoms, but 4.1% and 3.1% of WCs reported moderate and severe psychological stress, respectively.

## Factor associated with psychological stress among waste collectors

Three groups of factors were put into the multiple logistic regressions to examine their relationships with psychological stress among the participants. These groups were the frequency of exposure to occupational hazards (13 factors), work organization (3 factors), and personal characteristics (7 factors). The results were presented in Table 2.

Among occupational hazards, frequent exposure to cold and heat was inversely associated with the occurrence of stress symptoms (OR = 0.51 [95% CI = 0.26–0.99]; OR = 0.52 [95% CI = 0.28–0.98], respectively) while frequent exposure to hot/flammable objects increased the odds of stress by 2.41 (95% CI 1.29–4.52) among WCs. No significant associations between stress symptoms and exposure to other hazards at work were observed.

Regarding work organization characteristics, only work shifts were significantly associated with the likelihood of stress among study participants. WCs who worked at night were more likely to be stressed than those working the day shift or frequently changing shifts (OR = 1.82, 95% CI = 1.03–3.24).

A statistically significant difference was observed among WCs with different education levels and monthly incomes. WCs who had not completed high school were 1.83 times more

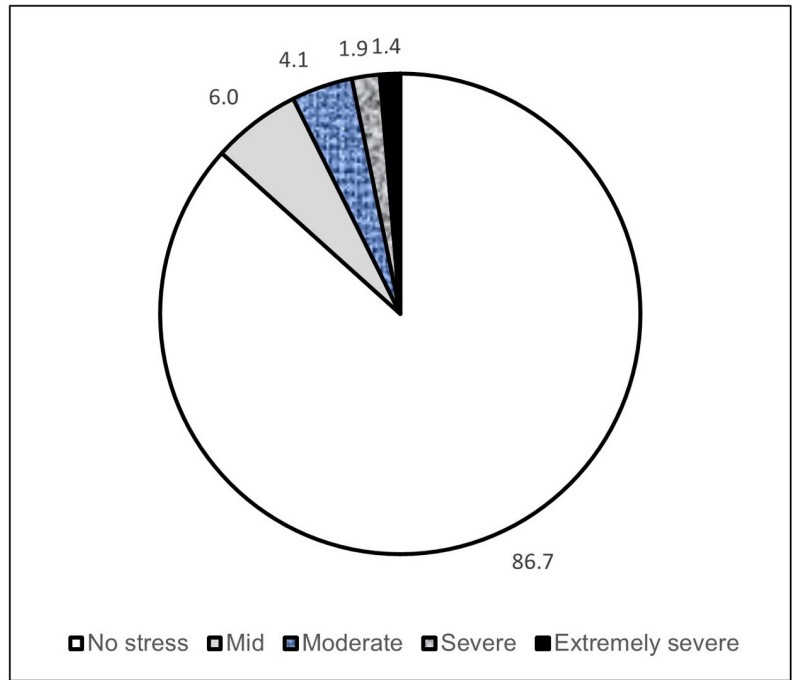

**Fig 1. Level of stress among waste collectors in this study (N = 802).**

**Table 2. Factors associated with psychological stress among waste collectors in this study.** Multivariate Logistics regression (N = 802).

| Variables | Subgroups | Whole sample | | Stress symptom | | Adjusted OR | 95%CI | |
|---|---|---|---|---|---|---|---|---|
| | | N = 802 | % | N = 107 | % | | Lower | Upper |
| **Frequency of exposure to occupational hazards** | | | | | | | | |
| Heat | Frequent | 465 | 58.0 | 59 | 12.7 | 0.51* | 0.26 | 0.99 |
| | Seasonal to none[a] | 337 | 42.0 | 48 | 14.2 | | | |
| Sunlight | Frequent | 378 | 47.1 | 56 | 14.8 | 1.75 | 0.90 | 3.43 |
| | Seasonal to none[a] | 424 | 52.9 | 51 | 12.0 | | | |
| Cold | Frequent | 227 | 28.3 | 29 | 12.8 | 0.52* | 0.28 | 0.98 |
| | Seasonal to none[a] | 575 | 71.7 | 78 | 13.6 | | | |
| Wetness | Frequent | 225 | 28.1 | 35 | 15.6 | 1.14 | 0.62 | 2.12 |
| | Seasonal to none[a] | 577 | 71.9 | 72 | 12.5 | | | |
| Dust | Frequent | 613 | 76.4 | 85 | 13.9 | 1.21 | 0.55 | 2.66 |
| | Seasonal to none[a] | 189 | 23.6 | 22 | 11.6 | | | |
| Toxic gas | Frequent | 393 | 49.0 | 82 | 20.9 | 1.03 | 0.62 | 1.73 |
| | Seasonal to none[a] | 409 | 51.0 | 25 | 6.1 | | | |
| Noise | Frequent | 575 | 71.7 | 41 | 7.1 | 1.21 | 0.57 | 2.54 |
| | Seasonal to none[a] | 227 | 28.3 | 66 | 29.1 | | | |
| Poor illumination | Frequent | 216 | 26.9 | 63 | 29.2 | 1.58 | 0.91 | 2.75 |
| | Seasonal to none[a] | 586 | 73.1 | 44 | 7.5 | | | |
| Flammable/ hot objects | Frequent | 131 | 16.3 | 31 | 23.7 | 2.41** | 1.29 | 4.52 |
| | Seasonal to none[a] | 671 | 83.7 | 76 | 11.3 | | | |
| Sharp object | Frequent | 259 | 32.3 | 43 | 16.6 | 0.88 | 0.51 | 1.50 |
| | Seasonal to none[a] | 543 | 67.7 | 64 | 11.8 | | | |
| Threat of mental violence | Yes | 21 | 2.6 | 7 | 33.3 | 1.26 | 0.71 | 2.23 |
| | Never[a] | 781 | 97.4 | 100 | 12.8 | | | |
| Threat of physical violence | Yes | 12 | 1.5 | 1 | 8.3 | 0.40 | 0.14 | 1.18 |
| | Never[a] | 790 | 98.5 | 106 | 13.4 | | | |
| Unpleasant smell | Frequent | 699 | 87.2 | 96 | 13.7 | 1.05 | 0.49 | 2.27 |
| | Seasonal to none[a] | 103 | 12.8 | 11 | 10.7 | | | |
| **Work organization** | | | | | | | | |
| Work shift in the last three months | Night shift | 584 | 72.8 | 89 | 15.2 | 1.82* | 1.03 | 3.24 |
| | Others[a] | 218 | 27.2 | 18 | 8.3 | | | |
| Work hours | > 8 hours/day | 159 | 19.8 | 16 | 10.1 | 0.66 | 0.36 | 1.21 |
| | ≤ 8 hours/day[a] | 643 | 80.7 | 91 | 14.2 | | | |
| Years of service | ≥15 years | 205 | 25.6 | 41 | 20.0 | 1.48 | 0.84 | 2.64 |
| | < 15 years[a] | 597 | 74.4 | 66 | 11.1 | | | |
| **Demographic characteristics and physical health** | | | | | | | | |
| Age | ≥39 | 321 | 40.0 | 55 | 17.1 | 1.14 | 0.65 | 1.99 |
| | <39[a] | 481 | 60.0 | 52 | 10.8 | | | |
| Marriage relationship | Others | 69 | 8.6 | 13 | 18.8 | 1.56 | 0.77 | 3.16 |
| | Married and live with a spouse[a] | 733 | 91.4 | 94 | 12.8 | | | |
| Gender | Male | 134 | 16.7 | 16 | 11.9 | 1.09 | 0.59 | 2.05 |
| | Female[a] | 668 | 83.3 | 91 | 13.6 | | | |
| Education | Below high school | 155 | 19.3 | 31 | 20.0 | 1.83* | 1.10 | 3.03 |
| | High school and above[a] | 647 | 80.6 | 76 | 11.8 | | | |
| Children | Not yet | 20 | 2.5 | 3 | 15.0 | 1.46 | 0.38 | 5.57 |
| | ≥ 1[a] | 782 | 97.5 | 104 | 13.3 | | | |

(*Continued*)

**Table 2.** (Continued)

| Variables | Subgroups | Whole sample | | Stress symptom | | Adjusted OR | 95%CI | |
| --- | --- | --- | --- | --- | --- | --- | --- | --- |
| | | N = 802 | % | N = 107 | % | | Lower | Upper |
| Monthly income | Insufficient | 413 | 51.5 | 73 | 17.7 | 1.99* | 1.22 | 3.24 |
| | Normal to sufficient[a] | 389 | 48.5 | 34 | 8.7 | | | |
| Physical functioning | < 95 | 490 | 61.1 | 81 | 16.5 | 0.40 | 0.14 | 1.18 |
| | ≥ 95[a] | 312 | 38.9 | 26 | 8.3 | | | |

Note:

[a] reference group,

* p<0.05;

** p<0.01;

CI = Confidence Interval; N = number of respondents; OR = odd ratio.

likely to report stress symptoms compared to those who had obtained at least a high-school degree (95% CI = 1.10–3.03, p < 0.05). Similarly, WCs who reported insufficient income had a risk of mental distress 1.98 times higher than WCs with a better financial condition (95% CI = 1.22–3.24, p < 0.05).

## Discussion

Stress symptoms were not prevalent among WCs. However, several WCs experienced moderate to extreme levels of stress. Working in a hot or cold work environment, exposure to hot/flammable objects, work shift, educational attainment and monthly income were among the factors significantly associated with stress among study participants.

### Prevalence of psychological stress among waste collectors

Although the mean score (standard deviation) of DASS-S among WCs in this study was moderate (9.7 ± 7.8), 4.1% and 3.1% of WCs showed moderate to severe levels of stress symptoms, respectively. Research on psychological stress among WCs has thus far been limited [4, 10–13]. Among them, only one study reported the prevalence of anxiety and depression among Japanese MSW incinerator workers, using the Japanese version of the Profile of Mood States Test [13]. Other studies investigated stressors [10, 11] or the psychological work environment using the Job Content Questionnaires [4, 12]. The higher prevalence of mental disorders in the Japanese study might be attributable to the study participants' occupation. Incinerator workers were aware and afraid of their occupational exposure to extremely hazardous agents such as dioxin which are released during the incineration process [13]. This knowledge elevated self-perceived mental distress among Japanese MSW workers. In addition, the Japanese study and the present study used different scales to measure other latent traits of mental disorders (anxiety, depression versus stress). Future studies should investigate a broader group of mental disorders to provide a comprehensive picture of the mental health of WCs, particularly in low- and middle-income countries where little support was available for their job of collecting waste. In addition, a low level of stress might be attributable to several demographic characteristics among WCs in this study. Many of them were married, lived with their spouse, and already had at least one child. Studies showed that being in a committed relationship improves the mental health of workers [27]. Moreover, WCs had good physical health, with a physical functioning mean score of 90.4 out of 100, and about 40% of WCs had physical functioning scores over 95. However, this phenomenon may not be representative of the WC population,

since injured or sick workers might temporarily or permanently be absent at the worksite at the time of the data collection visit and not participate in this study because their health condition was not suitable for the job which required physical strength for vigorous movement and heavy lifting.

## Factor associated with psychological stress among waste collectors

Multivariate logistics regression showed several significant associations between the occurrence of stress and working conditions and individual characteristics.

The outdoor nature of MSW collection puts WCs at risk of exposure to various climatic conditions. The sub-tropical climate of Northern Vietnam is characterized with by increasingly extreme weather events (extremely hot in the summer and unpleasantly cold in the winter) due to climate change [28] and rapid urbanization [29]. Exposure to extreme heat, coupled with the heavy manual labor required in the job, was likely to cause mental fatigue [6] or increase the risk of physical health problems such as musculoskeletal disorders [30], which positively correlated with workers' mental well-being [12]. However, in our study, WCs with frequent exposure to heat and cold had lower odds of psychological stress compared with those who reported occasional/ seasonal or no exposure. One possible reason was that WCs who were frequently exposed to a hot environment might be more acclimated to heat [31, 32], and thus able to avoid the health effects of heat. Moreover, studies also reported that cold temperatures reduced negative mental health outcomes among adults [33].

Categorization and sorting of MSW is not a common practice in Vietnam [21]; hence, MSW WCs are exposed to various waste types and hazards. However, in this study, only a significant association between the frequency of exposure to hot/flammable objects and stress symptoms occurred, with more frequent exposure resulting in a greater risk of psychological stress among WCs. The hot/flammable objects that WCs frequently had to dispose of were mainly beehive coal briquettes, which were still commonly used even in a large city like Hanoi [34]. This kind of fuel was used by both households and small street-food restaurants and was disposed of along with other waste. From houses, it was not much of a threat since the coal would turn cold overnight. However, the burning coal from the restaurants was carelessly thrown on the roadside for the WCs to clean up. Working in such conditions increased workers' fear of fires starting when the hot coal came in contact with other waste in the wheeled bin, or of getting burns themselves. It also took WCs more time to treat this type of waste before it could be put into the bin with other waste, adding to their already heavy labor of manually collecting MSW.

Regarding the higher odds of stress among WCs working the night shift (OR = 1.82, 95% CI = 1.03–3.24), at night, WCs were susceptible to sleep deprivation, harsher climatic conditions compared to daytime, and fatigue, which escalated their risk of psychological distress [4]. Long work hour was a typical work organization for WCs who manually collected waste in alleys, on the streets using wheeled containers [4]. In Vietnam, the average official work hours were 8 hours per shift [35]. Still, for night shift workers, their job only ended when all MSW had been transported to the processing plant by a garbage truck, which made several round trips between the processing plant and MSW gathering locations during each working shift. Sometime, the truck returned later than scheduled, extending the shift of WCs because they had to wait to load all MSW onto a vehicle before their shift ended. Working at night also increased the risk of occupational injury and accidents due to insufficient natural light, fatigue, and drowsiness [4, 7], elevating night workers' anxiety about their safety.

Regarding demographic characteristics, WCs with lower educational attainment and monthly income had higher risk of stress symptoms (OR = 1.83, 95% CI = 1.10–3.03, and

OR = 1.98, 95% CI = 1.22–3.24, respectively). The high percentage of high school completers in our study was significantly different from studies in other low and middle-income countries such as Iran [4] and Brazil [36]. This high level of education among WCs was attributable to the fact that this was a competitive state-owned company. Because the company in this study provided better working conditions, social support, and earnings than private companies, it also requires WCs to have better qualifications. This result for education level was consistent with that for Japanese incinerator workers, among whom education level was inversely correlated with anxiety and depression [13]. In a competitive, modern urbanized society, higher educational qualifications are essential to secure a better work position, which promises better benefits and income. Having a higher education level also promotes health literacy among WCs, resulting in their better adherence to protective equipment use and attention to occupational services provided by the company such as occupational health and safety education. Hence, WCs with higher education levels might have better health awareness and take better care of their health, including mental health. Regarding income, insufficient income had been regarded as an occupational stressor among WCs in several low and middle-income countries such as Ghana [10] and Chile [11]. However, the study in Ghana did not find a significant association between income and stress levels among electronic waste recycling workers. In our study, more than half of WCs reported that their current income was inadequate to support their family. Coupled with the physical demands of manual labor, worry about their family's financial condition might have a detrimental effect on WCs' stress level.

Although in this study, neither gender nor age/year of services were significantly associated with stress, future studies on similar topics could consider the effect of these individual characteristics. In epidemiological studies on self-perceived mental health, it has been reported that gender differences play a vital role since men and women tend to react differently to stress [37]. Age and work seniority are essential determinants of workers' health, particularly manual labor, since older age results in a decrease in physical strength, making older workers susceptible to physical and mental exhaustion. In addition, more years of working increases the risk of workers being exposed to occupational hazards, which eventually affects their physical and psychological health.

The current study had several limitations. First, interviews regarding psychological stress and work conditions could have a degree of bias because of limitations on WCs' recall ability and awareness of the study topic, or social-desirability bias because of the interviewer's presence. Workers' awareness of occupational hazard exposure might result in overestimating negative health outcomes or over-reporting severe conditions among study participants. Second, it is impossible to determine the cause of psychological stress with the cross-sectional study design in this study; future studies should utilize a more robust study design such as a cohort study or randomized controlled trial. Third, generalizability of the study results is limited to companies with similar work conditions, and caution should be taken when generalizing the results to other study populations with different characteristics such as different gender ratios or education level. However, this is the first paper reporting the prevalence of psychological stress and related factors among Vietnam waste collectors. It is also among very few publications on this topic worldwide.

## Conclusions

The prevalence of psychological stress among WCs was modest (13.4%), but 3.3% of WCs reported severe to extremely severe levels of stress. Factors related to lower odds of self-reported psychological stress included self-perceived frequent exposure to high and low temperatures in the working environment. Factors associated with the increased likelihood of

stress symptoms were frequent exposure to hot/flammable objects, working at night in the last three months, not completing high school education and reporting insufficient monthly income. The high percentage of workers with a severe level of psychological stress implies the need for mental health check-ups and treatment for participating WCs. Continuous training on occupational exposure to hazards, particularly climatic conditions, and prevention of health risks such as heat acclimation or response to heat stress/ stroke is beneficial for improving the well-being of WCs. Social support for vulnerable groups such as WCs with lower education level or financial difficulties is also helpful for WCs to concentrate on their job without worrying about security in life for themselves and their families. The improvement in WCs' health and performance, in turn, enhances the quality of the MSW management system in low and middle-income countries where manual operation is irreplaceable in the meantime. Future studies should continue the investigation of mental health among the broader population of WCs from different companies to provide more representative evidence on WCs' mental health and related factors in Vietnam.

## Supporting information

**S1 Data. Survey questionnaires.**
(DOCX)

**S2 Data. Survey database.**
(SAV)

## Acknowledgments

The authors would like to express our gratitude to the Hanoi Department of Science and Technology for their financial and technical support for this study. We were thankful for the URENCO management board and waste collectors who participated in the survey.

## Author Contributions

**Conceptualization:** Quynh Thuy Nguyen, Bich Ngoc Nguyen, Thuy Thi Thu Tran.

**Data curation:** Quynh Thuy Nguyen, Bang Van Nguyen, Ha Thi Thu Do, Van Thanh Nguyen, Thuy Thi Thu Tran.

**Formal analysis:** Thuy Thi Thu Tran.

**Funding acquisition:** Quynh Thuy Nguyen.

**Investigation:** Quynh Thuy Nguyen, Ha Thi Thu Do, Van Thanh Nguyen, Thuy Thi Thu Tran.

**Methodology:** Quynh Thuy Nguyen, Bich Ngoc Nguyen, Thuy Thi Thu Tran.

**Project administration:** Quynh Thuy Nguyen, Ha Thi Thu Do, Van Thanh Nguyen.

**Resources:** Bang Van Nguyen, Ha Thi Thu Do.

**Supervision:** Quynh Thuy Nguyen.

**Writing – original draft:** Son Thai Vu, Thuy Thi Thu Tran.

**Writing – review & editing:** Quynh Thuy Nguyen, Bang Van Nguyen, Bich Ngoc Nguyen, Thuy Thi Thu Tran.

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
