## [Decision Letter · Decision Letter 0]

4 May 2021

PONE-D-20-31502

Psychological stress and associated factors among municipal solid waste collectors in Hanoi, Vietnam: A cross-sectional study

PLOS ONE

Dear Dr. Bang,

Thank you for submitting your manuscript to PLOS ONE. After careful consideration, we feel that it has merit but does not fully meet PLOS ONE’s publication criteria as it currently stands. Therefore, we invite you to submit a revised version of the manuscript that addresses the points raised during the review process.(please see below).

We look forward to receiving your revised manuscript.

Kind regards,

Harald Gündel

Academic Editor

PLOS ONE

Journal Requirements:

2.We suggest you thoroughly copyedit your manuscript for language usage, spelling, and grammar. If you do not know anyone who can help you do this, you may wish to consider employing a professional scientific editing service.  

4.In your Data Availability statement, you have not specified where the minimal data set underlying the results described in your manuscript can be found. PLOS defines a study's minimal data set as the underlying data used to reach the conclusions drawn in the manuscript and any additional data required to replicate the reported study findings in their entirety. All PLOS journals require that the minimal data set be made fully available. For more information about our data policy, please see http://journals.plos.org/plosone/s/data-availability.

5. We note you have included a table to which you do not refer in the text of your manuscript. Please ensure that you refer to Table 1 in your text; if accepted, production will need this reference to link the reader to the Table.

Additional Editor Comments:

Dear Bang,

I´m sorry for the delay, but major difficulties to find reviewers as well as current Covid - related consequences (additional clinical and other work) played a role.

Best, Harald Gündel

Reviewers' comments:

Reviewer's Responses to Questions

**Comments to the Author**

1. Is the manuscript technically sound, and do the data support the conclusions?

Reviewer #1: Yes

2. Has the statistical analysis been performed appropriately and rigorously? 

Reviewer #1: Yes

3. Have the authors made all data underlying the findings in their manuscript fully available?

Reviewer #1: Yes

4. Is the manuscript presented in an intelligible fashion and written in standard English?

Reviewer #1: Yes

5. Review Comments to the Author

Reviewer #1: This study showed the prevalence of psychological stress and related factors among waste collectors in Hanoi. The data was very valuable and the analytic process was appropriate. I pointed out some concerns to be considered.

Major comments.

1.[Materials and methods] The section of "Study area" is not appropriate here. Making the section shorter and inserting it into LN 83 seems suitable. Shorter information is enough to explain the settings and context.

2. [Materials and methods] Authors conducted the structured interview. How many researchers engaged in it? Was there any protocol on how to conduct it? How did you standardize the interview?

3. Please separate Results and Discussion.

4. [Conclusion] The implication for further study might be valuable for potential readers.

Minor comments.

1.[Abstract] Please state the scale authors used (DASS-21) in the methods section.

2.[Abstract] Authors use a variety of words to indicate the outcome (eg. "mental stress" "self-reported psychological stress" "stress symptoms"). Please unify them into the same term.

3. (LN 128) Why is there missing data on critical variables by conducting the interview?

4. [Methods] Did the participant receive any reward?

5. (LN176) The work-shift variable is needed to add more information. It is a very important variable. The response option "Yes" is included partial shift? Or, indicates only the everyday shift.

6. [discussion] Can knowledge about occupational safety, protective equipment use, and safety education influence the findings?

7. [discussion] Can interview cause any bias? (eg, Social-desirability bias)

6. PLOS authors have the option to publish the peer review history of their article (what does this mean?). If published, this will include your full peer review and any attached files.

Reviewer #1: **Yes: **Natsu Sasaki

---

## [Author Response · Author response to Decision Letter 0]

29 May 2021

RESPONSE TO REVIEWER’S COMMENTS: 

Reviewer #1: This study showed the prevalence of psychological stress and related factors among waste collectors in Hanoi. The data was very valuable and the analytic process was appropriate. I pointed out some concerns to be considered.

Major comments.

1.[Materials and methods] The section of "Study area" is not appropriate here. Making the section shorter and inserting it into LN 83 seems suitable. Shorter information is enough to explain the settings and context.

RESPONSE: 

- The information on study area (Hanoi city) was moved to the end of Introduction and shortened accordingly as follow:

“Similar to other countries in the world, a metropolis like Hanoi, the capital of Vietnam, also faces the challenge of sustainable MSW management [3, 15, 16]. Rapid urbanization, lifestyle changes, and high population density in the city generate an enormous amount of MSW every day, which is collected manually by WCs [17]. Since 2018, the population of Hanoi has increased to over 7.5 million people and the population density to 2.239 people/km2 [18]; its residents resided in 30 district-level administrative units, towns, 584 communes, wards, and towns [19]. Hanoi is located in the tropical monsoon region; hence, the climate is divided into four distinct seasons throughout the year. Temperatures can reach 40°C in summer and below 5°C in winter [20]. 

According to the Hanoi Department of Natural Resources and Environment, in 2019, about 6,500 tons of domestic MSW was generated daily, of which the MSW of 12 urban districts and Son Tay town accounted for 53.9%, with a 99-100% collection rate. MSW in 17 suburban districts was less than half of the daily amount, but only 87 to 88% were appropriately collected for disposal [15]. The laborious workload, coupled with harsh working conditions such as extreme heat in the summer and low temperatures in the winter, in association with lack of mechanical support for heavy manual work, leads WCs to become physically and mentally exhausted. However, evidence regarding the risk of mental distress among WCs worldwide and particularly in Vietnam is limited. Therefore, this study aimed to examine the prevalence of psychological stress and related factors among WCs in Hanoi, Vietnam. Results from this study will contribute to evidence-based solutions for improving the well-being of WCs in Vietnam and countries with similar MSW systems.”

2. [Materials and methods] Authors conducted the structured interview. How many researchers engaged in it? Was there any protocol on how to conduct it? How did you standardize the interview?

RESPONSE: 

- Additional information had been provided in the Data collection section as follow:

“Ten individuals, including researchers and graduate students with experience in conducting structured interviews, collected the data. They were trained thoroughly by the principal investigator on how to conduct the interview, and each practiced on five WCs who belonged to another waste collection company and did not participate in this survey. The principal investigator regularly monitored and supervised the interviewers’ performance and progress directly on the WCs’ worksite.”

3. Please separate Results and Discussion.

RESPONSE: 

- Results and Discussion sections had been separated

4. [Conclusion] The implication for further study might be valuable for potential readers.

RESPONSE: 

- Implication for further study had been provided at the end of conclusion as follow:

“Future studies should continue the investigation of mental health among the broader population of WCs from different companies to provide more representative evidence on WCs’ mental health and related factors in Vietnam.”

Minor comments.

1.[Abstract] Please state the scale authors used (DASS-21) in the methods section.

RESPONSE: 

- The name of the scale DASS 21 had been put in the abstract

2.[Abstract] Authors use a variety of words to indicate the outcome (eg. "mental stress" "self-reported psychological stress" "stress symptoms"). Please unify them into the same term.

RESPONSE: 

- The term psychological stress was used. 

3. (LN 128) Why is there missing data on critical variables by conducting the interview?

RESPONSE: 

The participants refused to answer even after the interviewer explained details. This sentence was revised as follow:

“Fifteen questionnaires were removed because participants refused to provide information on critical variables. The final sample used for the analysis was 802 WCs (response rate of 89.7%).”

4. [Methods] Did the participant receive any reward?

RESPONSE: 

- Additional information was provided at the end of the Data collection section as follow:

“After the interview, participants received an herbal tea box as compensation for their participation regardless of their answers. The tea box was worth less than one USD.”

5. (LN176) The work-shift variable is needed to add more information. It is a very important variable. The response option "Yes" is included partial shift? Or, indicates only the everyday shift.

RESPONSE: 

- Additional information was provided in the Measure section as follow: 

“Work conditions were described in terms of two groups of variables: work organization and exposure to occupational hazards. Work organization variables included the number of work hours per shift (≤ 8 versus 9-12 hours/shift) and work-shift during the last three months (Night shift, from 6 p.m. to 2 a.m., Others [including WCs working the day shift from 5 a.m. to 12 a.m., or the afternoon shift from 1 p.m. to 8 p.m.], or Frequently changed shifts during the last three months).”

6. [discussion] Can knowledge about occupational safety, protective equipment use, and safety education influence the findings?

RESPONSE: 

- This information was included in the discussion as follow:

“Having a higher education level also promotes health literacy among WCs, resulting in their better adherence to protective equipment use and attention to occupational services provided by the company such as occupational health and safety education. Hence, WCs with higher education levels might have better health awareness and take better care of their health, including mental health.”

7. [discussion] Can interview cause any bias? (eg, Social-desirability bias)

RESPONSE: 

- This information was included in the limitation section as follow:

“First, interviews regarding psychological stress and work conditions could have a degree of bias because of limitations on WCs’ recall ability and awareness of the study topic, or social-desirability bias because of the interviewer’s presence”

---

## [Editor Report · Decision Letter 1]

28 Jun 2021

Psychological stress and associated factors among municipal solid waste collectors in Hanoi, Vietnam: A cross-sectional study

PONE-D-20-31502R1

Dear Dr. Bang Van Nguyen,

We’re pleased to inform you that your manuscript has been judged scientifically suitable for publication and will be formally accepted for publication once it meets all outstanding technical requirements.

Kind regards,

Harald Gündel

Academic Editor

PLOS ONE

Additional Editor Comments (optional):

Thanks for this careful response to issues raised by reviewer #1.

---

## [Editor Report · Acceptance letter]

1 Jul 2021

PONE-D-20-31502R1 

Psychological stress and associated factors among municipal solid waste collectors in Hanoi, Vietnam: A cross-sectional study 

Dear Dr. Nguyen:

I'm pleased to inform you that your manuscript has been deemed suitable for publication in PLOS ONE. Congratulations! Your manuscript is now with our production department. 

Kind regards, 

on behalf of

Dr. Harald Gündel 

Academic Editor

PLOS ONE